# Differentiation of Murine C2C12 Myoblasts Strongly Reduces the Effects of Myostatin on Intracellular Signaling

**DOI:** 10.3390/biom10050695

**Published:** 2020-04-30

**Authors:** Juulia H. Lautaoja, Satu Pekkala, Arja Pasternack, Mika Laitinen, Olli Ritvos, Juha J. Hulmi

**Affiliations:** 1Faculty of Sport and Health Sciences, Neuromuscular Research Center, University of Jyväskylä, 40014 Jyväskylä, Finland; satu.p.pekkala@jyu.fi (S.P.); juha.j.t.hulmi@jyu.fi (J.J.H.); 2Department of Physiology, Faculty of Medicine, University of Helsinki, 00290 Helsinki, Finland; arja.pasternack@helsinki.fi (A.P.); olli.ritvos@helsinki.fi (O.R.); 3Department of Medicine, Faculty of Medicine, University of Helsinki, 00029 Helsinki, Finland; mika.laitinen@helsinki.fi; 4Department of Medicine, Helsinki University Hospital, 00029 Helsinki, Finland

**Keywords:** coculture, follistatin, inflammation, MAPK, myotube, skeletal muscle, Smad, tumorkine

## Abstract

Alongside in vivo models, a simpler and more mechanistic approach is required to study the effects of myostatin on skeletal muscle because myostatin is an important negative regulator of muscle size. In this study, myostatin was administered to murine (C2C12) and human (CHQ) myoblasts and myotubes. Canonical and noncanonical signaling downstream to myostatin, related ligands, and their receptor were analyzed. The effects of tumorkines were analyzed after coculture of C2C12 and colon cancer-C26 cells. The effects of myostatin on canonical and noncanonical signaling were strongly reduced in C2C12 cells after differentiation. This may be explained by increased follistatin, an endogenous blocker of myostatin and altered expression of activin receptor ligands. In contrast, CHQ cells were equally responsive to myostatin, and follistatin remained unaltered. Both myostatin administration and the coculture stimulated pathways associated with inflammation, especially in C2C12 cells. In conclusion, the effects of myostatin on intracellular signaling may be cell line- or organism-specific, and C2C12 myotubes seem to be a nonoptimal in vitro model for investigating the effects of myostatin on canonical and noncanonical signaling in skeletal muscle. This may be due to altered expression of activin receptor ligands and their regulators during muscle cell differentiation.

## 1. Introduction

The role of myostatin (growth differentiation factor 8, GDF8), a member of the transforming growth factor-β (TGF-β) family, as a negative regulator of muscle size is well recognized (for review, see [1,2]). Myostatin is a secreted protein that is expressed mainly in the skeletal muscle and to a lesser extent in the cardiac muscle and adipose tissue [3,4]. Besides coordinating normal growth and development in a healthy state, abnormal myostatin regulation promotes muscle atrophy (for review, see [5]). Myostatin binding to the activin type 2A and 2B receptors (ACVR2A/B) inhibits anabolic processes and muscle regeneration as well as enhances catabolic signaling in the skeletal muscle [6,7,8,9]. However, blocking tumor-derived cytokines (“tumorkines”), such as activin receptor (ACVR2) ligands (e.g., activins, myostatin and GDF11 [4,10,11]), by ACVR2 antagonism has attenuated muscle atrophy and improved survival in experimental cancer [8,12]. In addition, the inhibition of myostatin alone or myostatin and GDF11 by neutralizing antibodies has attenuated cancer cachexia in vivo and muscle atrophy in vitro [13]. Further, inhibition of myostatin and/or activins by follistatin induces muscle hypertrophy in vivo [14]. Follistatin is a circulating endogenous protein that strongly increases muscle mass and is a natural ACVR2 ligand blocker [14,15]. Myostatin signaling is inhibited when follistatin binds the C-terminal dimer of the myostatin protein, thus preventing association with ACVR2A/B [15].

Alongside in vivo models, a simpler and more mechanistic in vitro approach is required to study the effects of ACVR2 ligands on skeletal muscle. Unfortunately, the physiological responses may differ depending on the cell lines and organisms used, and thus the selection of the cell line should be made based on the research question [16]. Mouse C2C12 muscle cells are the most commonly used cellular model to mimic skeletal muscle in vitro. Various studies reporting the effects of myostatin on C2C12 proliferation and differentiation exist [17,18,19,20,21]. Administration of myostatin to C2C12 myoblasts has been reported to promote canonical and noncanonical signaling pathways [19,21]. Firstly, the canonical pathway is activated by myostatin binding to ACVR2A/B, which further induces the activation of activin receptor type 1B (or ALK4) or TGF-β receptor type 1 (or ALK5) to phosphorylate transcription factor Smad2/3 [22,23,24,25]. Secondly, Smad2/3 forms a complex with Smad4, which translocates into the nucleus to regulate gene expression [22,23,24]. Furthermore, activation of the Smad2/3 pathway promotes skeletal muscle catabolism at least via the ubiquitin–proteasome system [8,26,27]. The noncanonical pathway is responsive to the same TGF-β-related proteins, such as myostatin, that are able to promote stress and inflammation-responsive mitogen-activated protein kinases (MAPKs), such as p38 MAPK (p38), stress-activated protein kinase/c-Jun N-terminal kinases (SAPK/JNKs), and extracellular signal-regulated kinases (ERKs) [22], in vitro in C2C12 cells [28,29,30] and at least ERK in vivo [29,31]. For example, myostatin-induced growth inhibition of the C2C12 myoblasts was associated with p38 phosphorylation [28], while Zhang et al. reported that myostatin enhanced interleukin-6 (IL-6) production via the p38 pathway has been associated with proteolysis [32]. The kinases SAPK/JNKs and ERKs, on the other hand, have been associated with suppression of C2C12 myotube differentiation after myostatin administration [29,30].

Muscle wasting during cancer increases systemic inflammation [33,34] and inflammatory tumorkines, such as IL-6, monocyte chemoattractant protein 1 (MCP-1), and regulated on activation, normal T cell expressed and secreted (RANTES), activate transcriptional factors, such as signal transducer and activator of transcription 3 (STAT3) and CCAAT-enhancer-binding protein β (C/EBPβ) [34,35,36,37]. As a result, pathways related to muscle atrophy and expression of acute phase response proteins in the inflamed muscle are activated [34,35,36,37]. Indeed, ACVR2 ligands and other tumorkines disrupt normal skeletal muscle homeostasis during cancer [38]. The effects of tumorkines on intercellular signaling have been studied by applying cancer cell medium (conditioned medium, CM) to another cell line [39,40,41,42]. However, the Transwell^®^ coculture system [43] is a more suitable approach to study tumorkine effects because it enables a constant supply of the tumor-derived factors from the upper compartment to another cell population in the lower compartment. Jackman et al. found that although the concentration of the tumokine called leukemia inhibitory factor was constantly higher in the colon 26 carcinoma (C26)-CM than in the lower compartment of the Transwell^®^ system, there was a benefit of the Transwell^®^ model in that the coculture induced a greater decrease in the C2C12 myotube diameter [41,44].

The aim of this study was to investigate whether an in vitro approach could be utilized as a tool to examine the direct effects of myostatin on canonical and noncanonical signaling in widely used murine C2C12 myoblasts and myotubes. To validate the functionality of the recombinant myostatin protein and to exclude the possibility of cell line-specific responsiveness to myostatin, we repeated the experiments with human CHQ myoblasts and myotubes derived from quadriceps muscle biopsy [45]. Finally, we cocultured the muscle and cancer cells to examine the effects of tumorkines, including ACVR2 ligands, on muscle cell homeostasis.

## 2. Materials and Methods

### 2.1. Cell Cultures

Mouse C2C12 myoblasts (American Type Culture Collection, ATCC, Manassas, VA, USA) were cultured in high glucose containing Dulbecco’s Modified Eagle growth medium (GM) (DMEM, #BE12-614F, Lonza, Basel, Switzerland) supplemented with 10% (*v*/*v*) fetal bovine serum (FBS, #10270, Gibco, Rockville, MD, USA), 100 U/mL penicillin and 100 µg/mL streptomycin (P/S, #15140, Gibco), and 2 mM l-glutamine (#17-605E, Lonza). For the experiments, myoblasts were seeded on 12-well or 6-well plates (Nunclon^TM^ Delta; Thermo Fisher Scientific, Waltham, MA, USA). When the myoblasts reached 95–100% confluence, the cells were rinsed with phosphate-buffered saline (PBS) and the GM was replaced by differentiation medium (DM) containing high glucose DMEM, 5% (*v*/*v*) FBS or 2% (*v*/*v*) horse serum (HS, 12449C, Sigma-Aldrich, St. Luis, MO, USA) as indicated, 100 U/mL and 100 µg/mL P/S and 2 mM l-glutamine to promote fusion into myotubes. Fresh DM was changed every second day. Representative images of the C2C12 myoblasts and myotubes differentiated with 5% FBS DM are shown in the Appendix A. Human CHQ cells, derived from quadriceps muscle biopsy of a 5-day-old infant [45], were donated by Dr. Vincent Mouyly and Dr. Eija Laakkonen. The immortalization of this cell line does not have effects on myogenic cascades or any other cellular processes except restraining senescence, which justifies the usage of the commercial immortalized C2C12 cells and primary nonimmortalized CHQ cells in the present study [46,47]. Furthermore, the thigh muscles of mice and human from which these cell lines are obtained can be functionally considered similar and thus comparable [48]. The CHQ cells were cultured in growth medium with 4:1 ratio of GlutaMAX (#61965, Gibco) and Medium 199 (#41150, Gibco) supplemented with 20% (*v*/*v*) FBS (#10270, Gibco) and 50 µg/mL gentamicin (#15750, Gibco). After reaching over 90% confluence, the differentiation of CHQ myotubes was induced by replacing GM to DM containing 4:1 GlutaMAX and Medium 199, 50 µg/mL gentamicin, and 10 µg/mL bovine insulin (#15500, Sigma-Aldrich) as previously described [49]. The CHQ DM was not replaced during the differentiation. All the muscle cell experiments independent of the cell line used were conducted at day 0–1 (myoblasts in GM) or at day 5–6 (myotubes in DM) post differentiation. Colon 26 carcinoma (C26) cells were a kind gift from Dr. Fabio Penna. The C26 cells were grown in the same GM as described for the C2C12 myoblasts. All the cell experiments were performed in a humidified environment at 37 °C and 5% CO_2_.

### 2.2. Production and Administration of Myostatin

Recombinant myostatin protein was produced in house as described earlier for its receptor [50]. Shortly, the myostatin pro-and mature domains were amplified by PCR (prodomain 5′-TGGTCCAGTGGATCTAAATGAG-3′ and 5′-CTTTTTGGTGTGTCTGTTAC-3′, mature domain 5′-GAAGGGATTTTGGTCTTGAC-3′ and 5′-TCATGAGCACCCACAGCG-3′), and the domains were subcloned into pEFIRES-P vector. Both domains are required for the proper folding of the latent TGF-β family member precursors [51,52], and we expect myostatin to have similar requirement. This construct was co-transfected to CHO-S cells, positive cells were selected with puromycin (Thermo Fisher Scientific), and protein was produced in CD OptiCHO medium (Gibco) supplemented with 2 mM l-glutamine and grown in suspension in an orbital shaker. The protein was purified with HisTrap excel column (GE Healthcare, Chicago, IL, USA), eluted with imidazole, dialyzed against PBS, and finally concentrated with an Amicon Ultra concentrator (10,000 MWCO, Millipore, Burlington, MA, USA). Most of the purified myostatin was confirmed by western blot to be in the mature form (data not shown). To determine the activity of the myostatin preparation, HepG2 cells (purchased from ATCC) were cultured in DMEM supplemented with 10% (*v*/*v*) fetal calf serum (FCS), 2 mM l-glutamine, 100 U/mL, and 100 µg/mL P/S, 1× nonessential amino acids (Sigma-Aldrich) and 1 mM sodium-pyruvate (Sigma-Aldrich) and transfected with CAGA luciferase reporter construct as earlier with modifications [53]. The cells were stimulated with the ligand for 18 h and luciferase activity was measured with luciferase assay reagent (Promega, Madison, WI, USA). Dose-dependent activity was detected at 10–100 ng/mL and the activity of 10 ng of myostatin was abolished dose-dependently by addition of two myostatin blockers, soluble ACVR2B and follistatin (6–600 ng/mL) (data not shown). Prior to the administration to the cells, the inactive promyostatin was heated at 95 °C for five minutes to activate the mature myostatin protein [54]. The activated mature myostatin is highly conserved among human and mouse (99% identical; one amino acid difference between these species), which justifies the usage of the same myostatin on mouse C2C12 and human CHQ cell lines [4]. Before the experiments, the wells were rinsed twice with PBS and fresh medium, depending on the differentiation state GM or DM with all supplements, was added. The final myostatin concentration used was selected to be 100 ng/mL based on our dose-dependent bioassay (data not shown) and previous reports [21,32]. The 2-h time-point was selected to show acute changes and the 24-h time-point to show more delayed or persistent changes based on previous studies [18,21,55]. All myostatin experiments (*N* = 3) were independently replicated (total *N* = 6 per group).

### 2.3. Transwell^®^ Method, Coculture of C2C12 and C26 Cells

The C2C12 cells were seeded on 12-well or 6-well Transwell® plates (Costar, Corning Incorporated, Corning, NY, USA) and grown as described above; myoblasts were maintained in GM while myotubes were differentiated with 5% FBS DM. The C26 cells were grown in C2C12 GM or acclimatized for 48 h to C2C12 DM prior to the combination of the cell lines as demonstrated previously [41]. The C26 cells were seeded on the Transwell^®^ inserts with 0.4 µm porous membrane (Costar, Corning Incorporated) and grown on a separate plate. The C26 inserts were approximately 80–90% confluent when the myoblasts were 100% confluent and the myotubes were differentiated for five days. On day 0 or 5 post C2C12 differentiation, the medium of C2C12 and C26 cells was removed, wells and inserts were rinsed with PBS, and fresh GM or DM was added to both upper and lower compartments. Then, C26 inserts were placed on C2C12 wells for 24-h coculture with myoblasts or myotubes. The time-point for the coculture experiment was chosen based on previous studies [41,56]. Coculture experiments (*N* = 3–4) were independently replicated once (total *N* = 6–8 per group).

### 2.4. SUnSET Method for the Analysis of Protein Synthesis, Protein Extraction, and Western Blotting

To measure the level of protein synthesis from C2C12 cells during coculture with C26 cells, the surface sensing of translation (SUnSET) method was used as demonstrated previously [28]. Briefly, puromycin was added to a final concentration of 1 µM as previously reported by us [29]. From this step forward, protein extraction was performed to all samples as follows. After the treatments, the cells were rinsed twice with cold PBS and lysed and scraped on ice in a buffer containing 20 mM HEPES (pH 7.4), 1 mM EDTA, 5 mM EGTA, 10 mM MgCl_2_, 100 mM β-glycerophosphate, 1 mM Na_3_VO_4_, 1 mM DTT, 1% TritonX-100 and supplemented with protease and phosphatase inhibitors (#1861280, Thermo Fisher Scientific). After 30 min, the homogenate was centrifuged for 5 min, 500× *g* at +4 °C. Bicinchoninic Acid (BCA) Protein Assay Kit (Pierce Biotechnology, Rockford, IL, USA) was used according to manufacturers’ protocol to measure the total protein content with an automated KoneLab analyzer (Thermo Fisher Scientific, Vantaa, Finland). Western blot analysis was conducted as previously described [57]. In brief, ~7 or 10 μg of protein (CHQ and C2C12, respectively) were separated by SDS-PAGE, transferred to PVDF membranes, blocked and incubated overnight with primary antibodies at 4 °C. Proteins were visualized by enhanced chemiluminescence (SuperSignal west femto maximum sensitivity substrate; Pierce Biotechnology) using a ChemiDoc MP device (Bio-Rad Laboratories, Hercules, CA, USA) and quantified with Image Lab software (version 6.0; Bio-Rad Laboratories). When stain free protein synthesis (puromycin-incorporated proteins) and ubiquitinated proteins were analyzed, the whole lane intensity was quantified. Stain free was used as a loading control and the results were normalized by dividing the intensity of the analyzed band by the intensity of the whole stain free lane. When two bands of one protein were detected, the bands were quantified together. Antibodies used in this study are presented in the Appendix A (Appendix A).

### 2.5. RNA Extraction, cDNA Synthesis, and Quantitative Real-Time PCR

To extract total RNA, the cells were rinsed twice with cold PBS and lysed with TRIreagent solution according to the manufacturer´s protocol (AM9738, Thermo Fisher Scientific). The synthesis of cDNA and quantitative real-time PCR (RT-qPCR) were performed as previously described [57]. The sequences of the primers used in the study are presented in Appendix A. Amplicon lengths of the amplifications using self-designed primers not published previously were analyzed and the lengths were as expected (Appendix A). Detector of all double stranded DNA, PicoGreen (Quant-iT™ PicoGreen™ dsDNA Assay Kit, Thermo Fisher Scientific), was used to normalize the RT-qPCR results of both cell lines according to manufacturer´s protocol because no stable housekeeping gene was detected. For the analysis of the RT-qPCR results, $2^{-Ct}$ values were normalized to the PicoGreen content. The PicoGreen standard curves of both cell lines are presented in Appendix A. *N* = 3–5 per group.

### 2.6. Multiplex Cytokine Assay

The conditioned medium of the C26 cells (C26-CM) for multiplex cytokine assay (Q-Plex Array 16-plex ELISA, Quansys Biosciences, Logan, UT, USA) was collected as previously described [42]. Shortly, 25 μL of undiluted, overnight serum-starved C26-CM was centrifuged and passed through a 0.22 µm filter. The cytokine assay was performed according to the manufacturer’s protocol. The lower limit of detection for MCP-1 was 3.40 pg/mL and 2.15 pg/mL for RANTES. The calibrator range of the assay for both was 4.12–3000 pg/mL.

### 2.7. Statistical Analyses

The data were tested for normality (Shapiro–Wilk test) and equality of variances (Levene´s test) using IBM SPSS Statistics version 24 for Windows (IBM SPSS Statistics, Chicago, IL, USA). For statistical evaluation, a two tailed paired or unpaired Student´s *t*-test or Mann Whitney U-test (IBM SPSS Statistics) were used when appropriate. The results are presented as means ± SEM. The level of significance was set at *p* < 0.05. Pooling of the follistatin protein content results was justified because myostatin administration had no effect on the protein in question.

## 3. Results

### 3.1. Myostatin Had a Larger Effect on the Canonical, Noncanonical, and Inflammatory Signaling in C2C12 Myoblasts than in Myotubes

The C2C12 myoblasts differentiated into myotubes very successfully within five days as only a few cells remained as myoblasts at this time-point (Appendix A). The administration of myostatin increased the phosphorylation of Smad3^Ser423/Ser425^ after 2, but not after 24 h in C2C12 myoblasts (*p* < 0.001, Figure 1a,f), and this response was substantially lower in myotubes (*p* < 0.01, Figure 1a,f). On the noncanonical pathway, the 2-h myostatin administration induced phosphorylation of all MAPKs, p38^Thr180/Tyr182^, SAPK/JNK1/2^Thr183/Tyr185^, and ERK1/2^Thr202/Tyr204^ in myoblasts (*p* < 0.01, *p* < 0.01, and *p* < 0.05, respectively, Figure 1b–d,f,g), but again, in myotubes this response was very small or lost completely (Figure 1b–d,f,g). The comparison of nontreated C2C12 myotubes to myoblasts showed a nonuniform effect of differentiation into myotubes on canonical and noncanonical signaling (Figure 1e–g). To exclude the effect of the C2C12 myotube differentiation protocol, we repeated the experiment by using another differentiation method (2% HS), demonstrating that the decreased responsiveness of the C2C12 cells to myostatin after differentiation is independent of the differentiation protocol (Appendix A).

The 2 h myostatin administration promoted typical inflammatory pathways in the C2C12 cells, shown by the elevated phosphorylation of STAT3^Tyr705^ in both myoblasts and myotubes (*p* < 0.001 and *p* < 0.01, respectively, Figure 2a,d) and phosphorylated C/EBPβ^Thr235^ in the myoblasts (*p* < 0.01, Figure 2b,d). When comparing the nontreated C2C12 myotubes to myoblasts, the contents of phosphorylated STAT3^Tyr705^ and total STAT3 were higher in myotubes when compared to myoblasts, whereas the opposite was true regarding the phosphorylated C/EBPβ^Thr235^ and total C/EBPβ (Figure 2c,d).

### 3.2. The Coculture with C26 Cells Had Minor Effects on Canonical and Noncanonical Signaling in C2C12 Cells, but Increased Inflammatory Signaling Similar to the Effect of Myostatin

We previously reported that the C26 tumors express ACVR2 ligands, such as myostatin [12], and the same cell line is used in the present study. We analyzed inflammatory cytokines from the C26-CM using multiplex ELISA and found one highly (>1000 pg/mL) secreted inflammatory cytokine, MCP-1 (>3500 pg/mL), and one moderately (>100 pg/mL) secreted mediator, RANTES (>430 pg/mL), as shown by others [58]. MCP-1 and RANTES were also elevated in vivo in C26 tumor-bearing mice as we previously reported [12]. To analyze whether C26-derived tumorkines would also have more robust effects on myoblasts than they did on myotubes, we cocultured C2C12 cells in the Transwell^®^ system with C26 cancer cells for 24 h. Interestingly, rather than activating canonical and noncanonical pathways like myostatin administration, the 24-h coculture decreased the phosphorylation of Smad3^Ser423/Ser425^ in both myoblasts and myotubes (*p* < 0.05, Figure 3a,d), while the phosphorylation of p38^Thr180/Tyr182^ remained unaltered in myoblasts and decreased in myotubes (*p* < 0.05, Figure 3b,e). Furthermore, after the 24-h coculture, no changes were observed in protein synthesis or in ubiquitinated proteins in C2C12 cells (Appendix A). Additionally, the downstream mediators of the mechanistic target of rapamycin (mTOR) complex 1, phosphorylated S6K1/2^Thr389^ and ribosomal protein S6^Ser240/Ser244^, remained also unaltered (Appendix A). The phosphorylation of Akt^Ser473^ downstream to mTOR complex 2 [59] tended to decrease in myoblasts when cocultured with C26 cells (*p* = 0.067, Figure 3c–e).

Similar to myostatin administration, the phosphorylation of STAT3^Tyr705^ and C/EBPβ^Thr235^ increased in both myoblasts (*p* < 0.001 and *p* < 0.05, respectively, Figure 4a–c) and myotubes (*p* < 0.01, Figure 4a–c) after the 24-h coculture with C26 cells.

### 3.3. Lower Myostatin Responsiveness in C2C12 Myotubes Was Associated with Altered Gene Expression of Myostatin Regulators

The distinct response of the C2C12 myoblasts and myotubes to myostatin suggested that the differentiation might have affected the regulation of the myostatin signaling pathway. Therefore, we analyzed the transcriptional level of the TGF-β family members and their blocker, follistatin [15], from myoblasts and myotubes. Compared to myoblasts, follistatin mRNA was increased by 20-fold (*Follistatin*, *p* < 0.05, Figure 5a) in myotubes with no changes in ACVR2B mRNA (*Acvr2b*, Figure 5b). Similar to follistatin, an increase in the mRNA of endogenous GDF11 (*Gdf11*, *p* < 0.001, Figure 5c), activin A *(InhibinβA, p* < 0.001, Figure 5d) and myostatin *(Gdf8, p* < 0.001, Figure 5e) were observed. The differentiation protocol again did not influence the results (Appendix A and Figure 5), with the exception of the unaltered myostatin level after 2% HS differentiation protocol (Appendix A).

Next, to validate the obtained follistatin mRNA result, we analyzed the protein content of follistatin from C2C12 cells. In agreement with the transcriptional level, follistatin protein content was increased by 2-fold in myotubes in comparison with myoblasts (*p* < 0.001, Figure 6a,b). Myostatin administration had no effect on the follistatin protein content (Figure 6a,b). Again, the effect was independent of the differentiation method (Appendix A).

### 3.4. Human Skeletal Myotubes Are Responsive to Myostatin

We replicated the myostatin experiment using human CHQ myoblasts and myotubes to analyze whether the reduced responsiveness of the canonical and noncanonical signaling pathways to myostatin was cell-line-specific. Independent of the differentiation stage, myostatin administration to human CHQ cells increased canonical and noncanonical signaling (Figure 7a–d,f–h). In contrast to C2C12 cells with a strong and fast response in inflammatory signaling (Figure 2a,b,d), myostatin administration to CHQ cells resulted in only a minor increase in the phosphorylated STAT3^Tyr705^ in myoblasts occurring after 24 h (Figure 8a,b,d). The differentiation process of CHQ myoblasts into myotubes (Figure 7e–h) was affected in a different manner than what was shown earlier in C2C12 cells (Figure 1e–g). More specifically, in C2C12 myotubes phosphorylated p38 was increased, while other phosphorylated MAPKs were decreased (Figure 1e–g), and both phosphorylated and total C/EBPβ were strongly diminished (Figure 2c,d). In contrast, in CHQ myotubes, phosphorylated p38 was decreased and most of the other phosphorylated or total MAPKs were increased (Figure 7e–h), while phosphorylated C/EBPβ was unaffected by the differentiation and total C/EBPβ was increased (Figure 8c,d). Phosphorylated STAT3 increased after differentiation (Figure 2c,d and Figure 8c,d), while Smad3 was nonresponsive to differentiation in both cell lines (Figure 1e–g and Figure 7e–h).

In contrast to C2C12 myotubes, the differentiation of CHQ myotubes had a smaller effect on the ACVR2 ligands and follistatin. The mRNA of follistatin (*FOLLISTATIN*), ACVR2B (*ACVR2B*), and GDF11 (*GDF11*) remained unaltered in CHQ myoblasts and myotubes, while activin A (*INHIBINβA*, *p* < 0.01) and myostatin (*GDF8*, *p* = 0.073) decreased (Figure 9a–e). Unlike in C2C12 cells, follistatin protein content remained unaltered among CHQ differentiation stages but was similarly unaffected by myostatin administration (Figure 9f,g). These results highlight the differences between C2C12 and CHQ skeletal muscle cells and their distinct ability to regulate endogenous ACVR2 ligands and their blocker, follistatin.

## 4. Discussion

In this study, we report that differentiation of C2C12 myoblasts into myotubes strongly reduced the effects of myostatin on canonical (Smad3) and noncanonical (MAPKs) signaling. Similarly, myostatin-induced phosphorylation of the inflammatory signaling proteins was greater in myoblasts than in myotubes. The inflammatory pathways, however, were similarly promoted in myoblasts and myotubes after coculture of the C2C12 cells with C26 cancer cells. This suggests that the general responsiveness to inflammation-regulating tumorkines is not reduced after differentiation. The reduced effects of myostatin in C2C12 myotubes was accompanied by enhanced expression of follistatin, an inhibitor of the TGF-β family members, and increased mRNA expression of ACVR2 ligands in C2C12 myotubes in comparison to myoblasts. Unlike in murine C2C12 cells, myostatin-induced changes in human CHQ skeletal muscle cells were mostly independent of the differentiation stage. Moreover, unlike in C2C12 cells, no changes in follistatin expression were observed while the mRNA expression of some ACVR2 ligands was even decreased in myotubes when compared to myoblasts. The reason for the lower responsiveness of the C2C12 myotubes to ACVR2 ligand myostatin may be related to the elevated follistatin content in comparison to myoblasts, because follistatin is a very strong blocker of myostatin [15]. In line with our findings, a microarray demonstrated a 50-fold increase in follistatin mRNA in C2C12 myotubes in comparison to myoblasts [60]. Rossi et al. reported that follistatin mRNA increased during differentiation in C2C12 cells, while ACVR2B mRNA expression remained unaltered [61]. Higher follistatin content in myotubes may be explained by the fact that follistatin has an important role in the myogenic differentiation and force generation for contractions in C2C12 myotubes [62]. These studies highlight the vital role of follistatin in the functionality and contractility of the C2C12 cells, which are more relevant in the myotubes than in the myoblasts.

Another possible explanation for the reduced responsiveness to myostatin in C2C12 myotubes may be the simultaneous increase in the mRNA expression of myostatin, GDF11, and activin A. Furthermore, others have previously shown that myostatin mRNA increased during C2C12 myoblast differentiation [61,63,64], and suggested that endogenous myostatin has a paracrine function as a regulator of myoblast proliferation and cell survival [63]. Similarly, activin A, another regulator of myogenesis, has also been related to the inhibition of proliferation and differentiation [23]. In the present study, we further expand these results by showing that GDF11 mRNA increased in differentiated C2C12 myotubes. Unlike in murine cells, the differentiation of the human CHQ muscle cells did not elevate the gene expression or the protein content of follistatin or mRNA expression of ACVR2 ligands. Therefore, we speculate that this difference in ACVR2 ligands and their regulator, follistatin, between these cell models may explain different responses to myostatin. The metabolic differences of mouse C2C12, rat L6 and human skeletal muscle cell lines have been shown, and thus cell-line selection should be made based on the research question [16]. Similarly, our results suggest that C2C12 myotubes may not be the most suitable in vitro model to investigate the effects of myostatin on canonical and noncanonical signaling in skeletal muscle. However, the detailed mechanisms for the reduced myostatin response in C2C12 myotubes require further examination.

Canonical Smad signaling is a key pathway of TGF-β signaling [65]. In vivo studies have shown that myostatin induces muscle atrophy via Smad3 [66,67,68]. In line with our in vitro findings, Yuzawa et al. reported a greater increase in the phosphorylation of Smads after shorter (30 min) than longer (6–12 h) treatment with 300 ng/mL of myostatin in C2C12 cells [69]. Our results show that canonical signaling was increased in the C2C12 myoblasts, while this response was blunted in the myotubes. The noncanonical signaling pathway (MAPKs) has also been shown to play a role in muscle atrophy induced by the ACVR2 ligands [24,70]. We observed that myostatin administration increased the phosphorylation of all studied MAPKs in C2C12 myoblasts, while only p38 phosphorylation increased in myotubes. Therefore, similar to the canonical pathway, C2C12 myotubes lost most of their noncanonical responsiveness to myostatin during differentiation. It is possible in the C2C12 cell line that the presence of some myoblasts in the myotube culture could at least partly explain the minor remaining responsiveness to myostatin in the latter (Appendix A). Our in vitro results are comparable to in vivo situation because the noncanonical signaling pathway is also responsive to myostatin in muscles in vivo. For instance, an increase in the phosphorylation of ERK1/2 in myostatin-administered mice has been reported [29], while blocking of the endogenous myostatin and activins decreased MAPK-signaling in murine muscle [50]. In contrast to our murine in vitro results, but in agreement with previous studies conducted using commercial human muscle cell lines [28,67], we observed that administration of myostatin enhanced both canonical and noncanonical signaling pathways in human CHQ myoblasts and myotubes. Interestingly, another difference between these cell lines was that differentiation of the C2C12 and CHQ myoblasts into myotubes resulted in opposite effects on MAPK-signaling, while Smad3-signaling remained unaffected after differentiation in both cell lines. However, myostatin had only a minor effect on inflammatory pathways in CHQ cells in contrast to C2C12 cells when both cell lines responded to differentiation by increased STAT-signaling and phosphorylated C/EBPβ decreased strongly only in C2C12 cells. Moreover, the effect of myostatin on inflammatory signaling was small in CHQ cells in contrast to C2C12 cells. The results suggest that the effects of myostatin may be more prominent in cell lines other than murine C2C12, and this should be taken into consideration when choosing an in vitro model for muscle research. However, in the future, time-series of the effects of myostatin on canonical and noncanonical signaling in the skeletal muscle cells should be conducted.

In recent years, a better understanding of the tissue crosstalk in diseases causing muscle atrophy, such as cancer-associated cachexia, has become important [71]. Cancers can be accompanied by tumorkine-induced inflammation [72]. Similarly, we observed in vitro that both the myostatin administration and the cancer cell coculture stimulated inflammatory pathways in C2C12 myoblasts and myotubes by increasing the phosphorylation of STAT3 and C/EBPβ, markers of increased inflammatory response [35]. However, although STAT3 has been reported to promote catabolic processes in the muscle during inflammation-induced cachexia [34], in C2C12 cells, STAT3 has been shown to regulate both proliferation and differentiation [73,74] and indeed p-STAT3 and STAT3 were elevated in myotubes when compared to myoblasts in the present study. This highlights the complexity of the interpretation of STAT3 signaling in different physiological conditions. Similarly to our study, Ding et al. reported that myostatin administration increased the phosphorylation of C/EBPβ in C2C12 myotubes [55], and they further linked p38β-regulated C/EBPβ to catabolic processes [75]. However, the present study suggests that the C26 cell-derived tumorkines promote inflammation rather than antianabolic or catabolic signaling because no changes were observed in protein synthesis, ubiquitinated proteins, or in mTOR signaling. Similarly to STAT3, C/EBPβ may have a role in the regulation of C2C12 myoblast differentiation. Mancini et al. demonstrated that C2C12 myoblasts contained detectable levels of C/EBPβ in the early phase of differentiation into adipocytes, while C/EBPβ was not observed during differentiation into myotubes [76]. This suggests that C/EBPβ may regulate the selection of the differentiation pathway in C2C12 cells, which could explain why C/EBPβ was barely detected in myotubes in the present study. To summarize, our results indicate that the C26 cell-derived tumorkines, although they have been shown to decrease myotube size after 24-h coculture [41], seemed to have only minor effects on antianabolic and catabolic signaling at this time-point. Furthermore, time-series experiments on the tumorkine secretion kinetics during coculture in various muscle cell lines should be conducted in the future.

## 5. Conclusions

In this study, we investigated whether an in vitro muscle cell approach could be utilized as a tool to study the direct effects of myostatin and tumorkines on myoblasts and myotubes. The results demonstrate that the canonical and noncanonical responses to myostatin were dependent on the differentiation stage of the C2C12 cells as well as on the signaling pathway studied. The reduced myostatin responsiveness of the C2C12 myotubes could possibly be due to the endogenous regulation of the TGF-β family members and their regulators during differentiation. Importantly, the results demonstrate that the decreased effects of myostatin on canonical and noncanonical signaling were not a universal phenomenon in cell models, because the human muscle cells restrained the responsiveness to myostatin. Therefore, according to our results, C2C12 myotubes are not a recommended cell line to study the effects of myostatin and possibly other ACVR2 ligands.

## Figures and Tables

**Figure 1 biomolecules-10-00695-f001:**
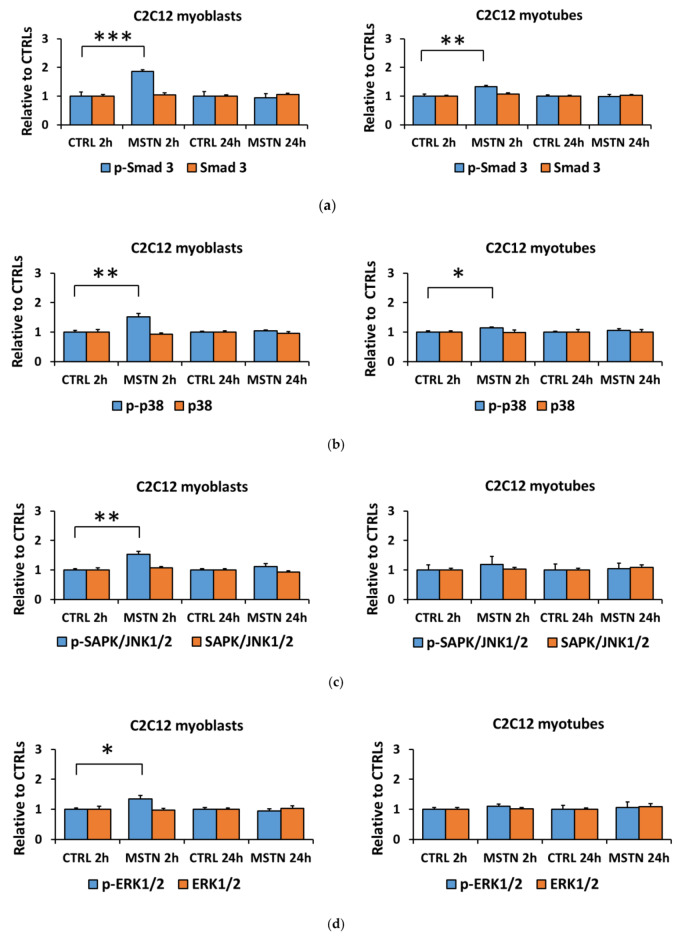
Myostatin-induced changes in the canonical (Smad3) and noncanonical (MAPKs) pathways were greater in C2C12 myoblasts than they were in myotubes. (**a**) Phosphorylated Smad3^Ser423/425^ and total Smad3 in myoblasts and myotubes. (**b**) Phosphorylated p38^Thr180/Tyr182^ and total p38 in myoblasts and myotubes. (**c**) Phosphorylated SAPK/JNK1/2^Thr183/Tyr185^ and total SAPK/JNK1/2 in myoblast and myotubes. (**d**) Phosphorylated ERK1/2^Thr202/Tyr204^ and total ERK1/2 in myoblasts and myotubes. In the figures, the values are presented as normalized to CTRL = 1. (**e**) Nontreated CTRL myoblasts and myotubes of the 2-h and 24-h time-points were pooled and the values are presented as normalized to myoblasts = 1. (**f**,**g**) Representative blots. In A–D, *N* = 6 per group. In E, *N* = 12 per group. *, **, and *** = *p* < 0.05, *p* < 0.01, and *p* < 0.001, respectively. CTRL (-) = control group, MSTN (+) = myostatin group.

**Figure 2 biomolecules-10-00695-f002:**
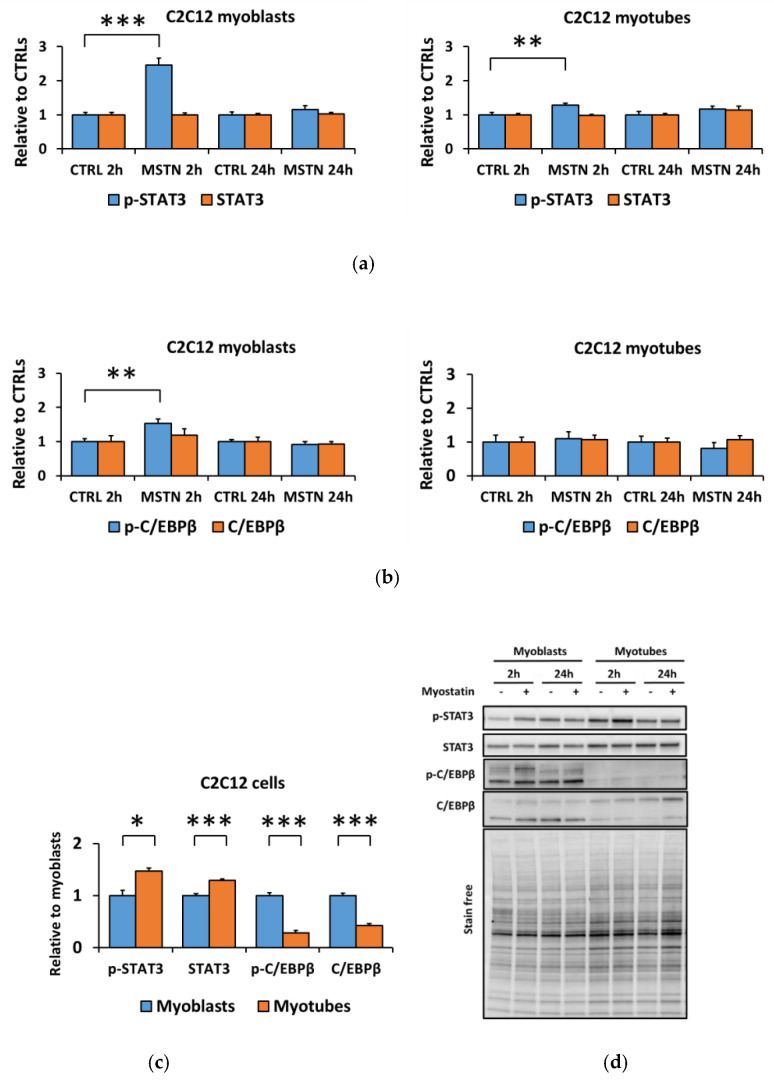
Myostatin administration on C2C12 cells increased inflammatory signaling in C2C12 myoblasts, whereas in myotubes the effect was very small. (**a**) Phosphorylated STAT3^Tyr705^ and total STAT3 in myoblasts and myotubes. (**b**) Phosphorylated C/EBPβ^Thr235^ and total C/EBPβ in myoblasts and myotubes. In the figures, the values are presented as normalized to CTRL = 1. (**c**) Nontreated CTRL myoblasts and myotubes of the 2-h and 24-h time-points were pooled, and the values are presented as normalized to myoblasts = 1. (**d**) Representative blots. In A–B, *N* = 6 per group. In C, *N* = 12 per group. *, **, and *** = *p* < 0.05, *p* < 0.01, and *p* < 0.001, respectively. CTRL (-) = control group, MSTN (+) = myostatin group.

**Figure 3 biomolecules-10-00695-f003:**
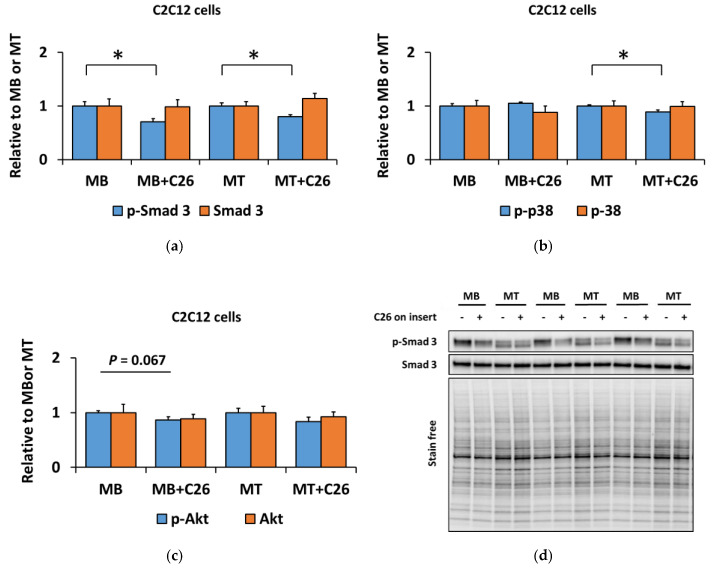
The 24-h coculture of C2C12 myoblasts (MB) and myotubes (MT) with C26 cells suppressed canonical and noncanonical signaling independent of the differentiation stage. (**a**) Phosphorylated Smad3^Ser423/425^ and total Smad3. (**b**) Phosphorylated p38^Thr180/Tyr182^ and total p38. (**c**) Phosphorylated Akt^Ser473^ and total Akt. In the figures, MB or MT with empty insert (-) are set as one and compared with MB or MT with C26 cells on the insert ((+), MB+C26 and MT+C26, respectively). (**d**,**e**) Representative blots. *N* = 6–8 per group. * = *p* < 0.05.

**Figure 4 biomolecules-10-00695-f004:**
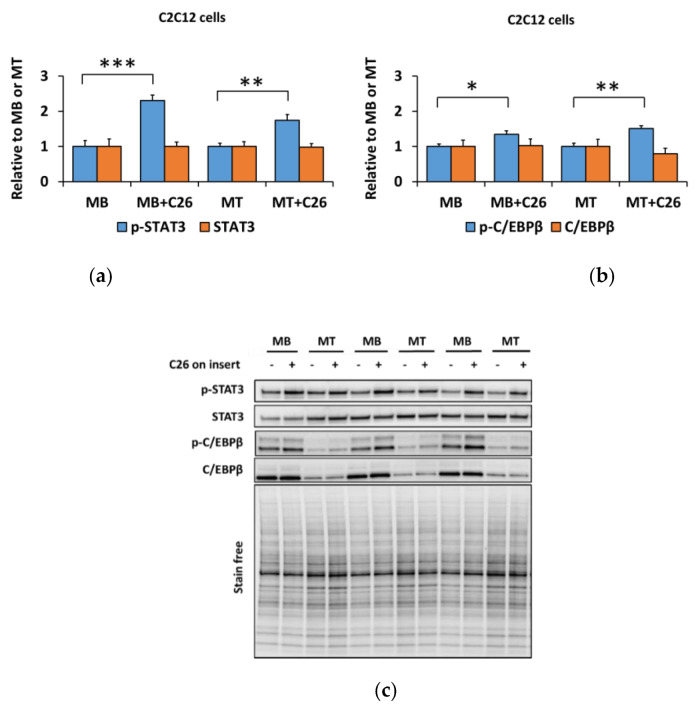
The 24-h coculture of C2C12 myoblasts (MB) and myotubes (MT) with C26 cells promotes inflammatory response independent of the differentiation stage. (**a**) Phosphorylated STAT3^Tyr705^ and total STAT3. (**b**) Phosphorylated C/EBPβ^Thr235^ and total C/EBPβ. (**c**) Representative blots. In the figures, MB or MT with empty insert (-) are set as one and compared with MB or MT with C26 cells on the insert ((+), MB+C26 and MT+C26, respectively). *N* = 6–8 per group. *, **, and *** = *p* < 0.05, *p* < 0.01, and *p* < 0.01, respectively.

**Figure 5 biomolecules-10-00695-f005:**
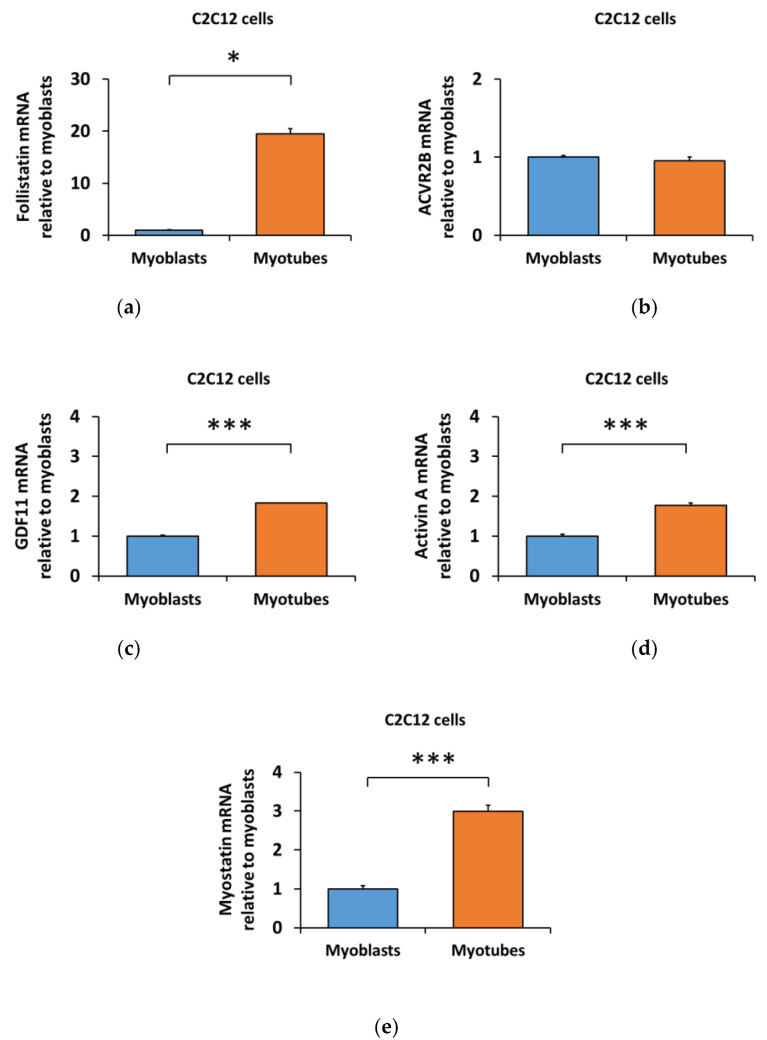
Differentiation of the C2C12 cells increased the mRNA level of (**a**) follistatin, while (**b**) ACVR2B remained unaltered. Of the ACVR2 ligands, (**c**) GDF11, (**d**) activin A, and (**e**) myostatin mRNAs were increased in myotubes in comparison to myoblasts. In the figures, the values are presented as normalized to myoblasts = 1. *N* = 3 per group. *, *** = *p* < 0.05 and *p* < 0.001, respectively.

**Figure 6 biomolecules-10-00695-f006:**
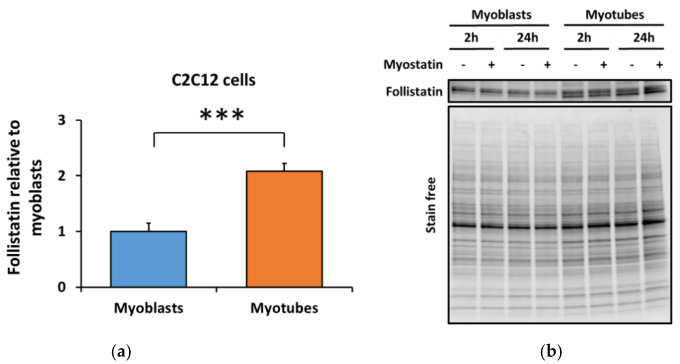
Differentiation of the C2C12 myoblasts into myotubes increased (**a**) follistatin protein content. In the figure, the values are presented as normalized to myoblasts = 1. (**b**) Representative blot. *N* = 12 per group as control (-) and myostatin (+) samples were pooled due to lack of myostatin effect. *** = *p* < 0.001.

**Figure 7 biomolecules-10-00695-f007:**
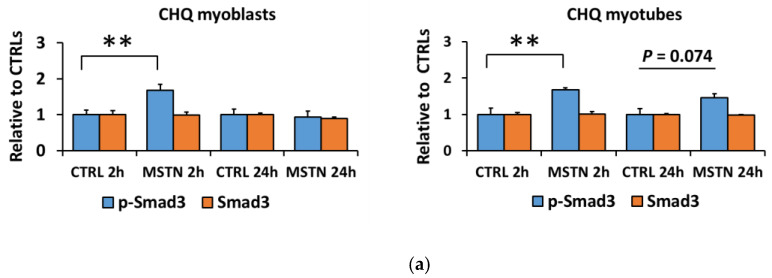
Myostatin-induced changes in the canonical (Smad3) and noncanonical (MAPKs) pathways were similar in CHQ myoblasts and myotubes**.** (**a**) Phosphorylated Smad3^Ser423/425^ and total Smad3 in myoblasts and myotubes. (**b**) Phosphorylated p38^Thr180/Tyr182^ and total p38 in myoblasts and myotubes. (**c**) Phosphorylated SAPK/JNK1/2^Thr183/Tyr185^ and total SAPK/JNK1/2 in myoblasts and myotubes. (**d**) Phosphorylated ERK1/2^Thr202/Tyr204^ and total ERK1/2 in myoblasts and myotubes. In the figures, the values are presented as normalized to CTRL = 1. (**e**) Nontreated CTRL myoblasts and myotubes of the 2-h and 24-h time-points were pooled and the values are presented as normalized to myoblasts = 1. (**f**–**h**) Representative blots. *N* = 6 per group. *, **, and *** = *p* < 0.05, *p* < 0.01, and *p* > 0.001, respectively. CTRL (-) = control group, MSTN (+) = myostatin group.

**Figure 8 biomolecules-10-00695-f008:**
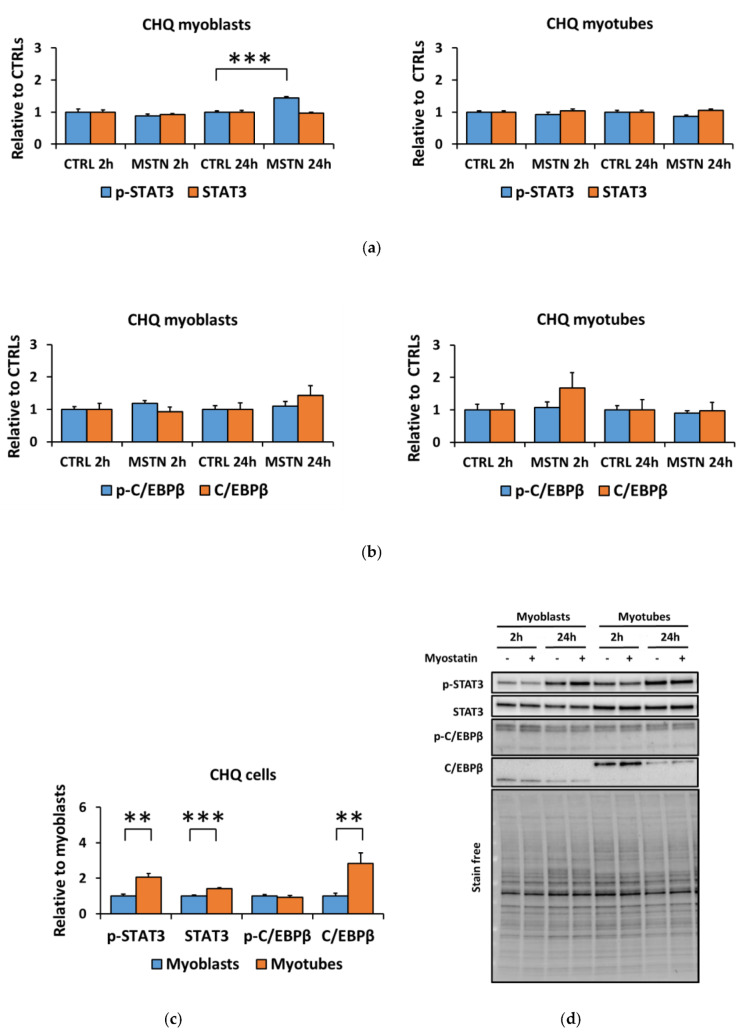
Myostatin administration had a minor impact on inflammatory signaling in CHQ cells. (**a**) Phosphorylated STAT3^Tyr705^ and total STAT3 in myoblasts and myotubes. (**b**) Phosphorylated C/EBPβ^Thr235^ and total C/EBPβ in myoblasts and myotubes. In the figures, the values are presented as normalized to CTRL = 1. (**c**) Nontreated CTRL myoblasts and myotubes of the 2-h and 24-h time-points were pooled and the values are presented as normalized to myoblasts = 1. (**d**) Representative blots. In A–B, *N* = 6 per group. In C, *N* = 12 per group. ** and *** = *p* < 0.01 and *p* < 0.001, respectively. CTRL (-) = control group, MSTN (+) = myostatin group.

**Figure 9 biomolecules-10-00695-f009:**
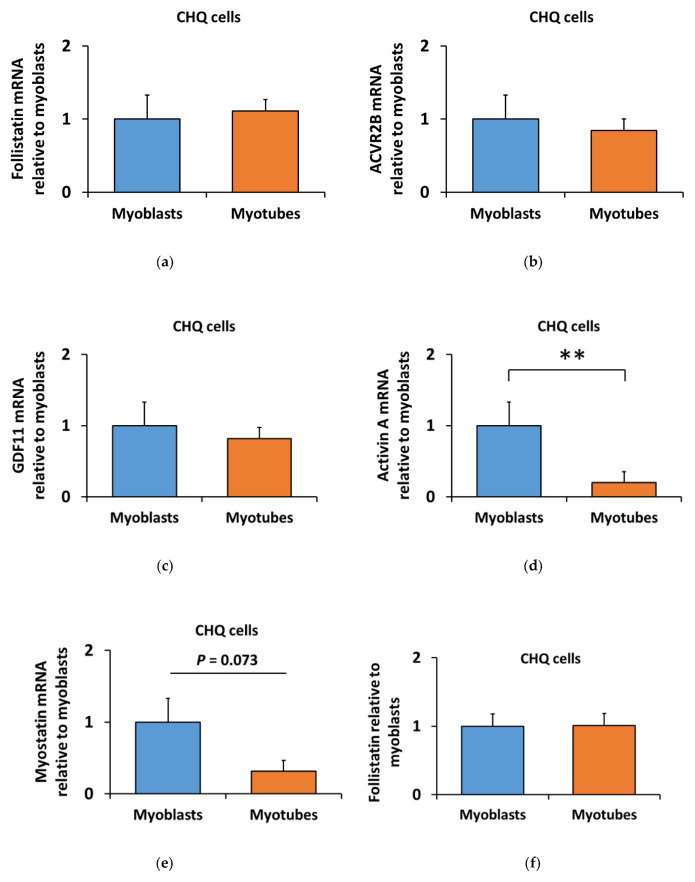
Differentiation of the CHQ cells did not alter the mRNA level of (**a**) follistatin, (**b**) ACVR2B and (**c**) GDF11. Of the ACVR2 ligands, the mRNA of (**d**) activin A and (**e**) myostatin were decreased in CHQ myotubes in comparison to myoblasts. (**f**) Differentiation of CHQ myoblasts into myotubes had no effect on follistatin protein content. (**g**) Representative blot. In the figures, the values are presented as normalized to myoblasts = 1. In A–E, *N* = 4–5 per group. In F, *N* = 12 per group as control (-) and myostatin (+) samples were pooled due to the lack of myostatin effect. ** = *p* < 0.01.

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
