# Peer review of "Differentiation of Murine C2C12 Myoblasts Strongly Reduces the Effects of Myostatin on Intracellular Signaling"

_biomolecules, 2020, doi:10.3390/biom10050695_

Round 1

Reviewer 1 Report

  1. There are some points need to be addressed. Figure 1f and g are the western blots showing the quantified results from 1a-e; however, the explanations for these two figures in the content is still needed. As well as 2d, 3d/e, 4c, 8c/d and 9g. Please specify which panel is referring to in figure 6 and 7 in the content. Every figures should be explained in the content, even just one sentence.

  1. In Figure 2c, although the result observed from myostatin effect is independent of the differentiation process, the p-STAT3 is increased in MT whereas p-C/EBPb is less in MT; since these two are inflammatory factors, what could be the possible reasons result in this opposite effect?

  1. There is no supplemental data attached? The link provided in the manuscript is malfunctioned.

Overall, this revised version is much clear and well-written, not only the language, but also logic thinking.

Author Response

Please see attachement.

Reviewer 2 Report

The revision of Lautaoja et al. "Differentiation of murine C2C12 myoblasts strongly 2 reduces the effects of myostatin on intracellular 3 signaling" has been largely satisfactory. The discussion of the expanded figure 7 turns out to be very poor. Therefore, these new results should be presented in more detail in the "Results" chapter (3.4).

Author Response

Please see attachement.

This manuscript is a resubmission of an earlier submission. The following is a list of the peer review reports and author responses from that submission.

Round 1

Reviewer 1 Report

Brief Summary: Lautaoja et al was investigating whether the current in vitro model, murine C2C12 myoblast cell line, is suitable to examine the direct effect of myostatin on canonical and non-canonical pathways. Besides, cancer-associated cachexia that results in muscle atrophy were also studied by evaluating tumor-derived cytokines analysis.

By comparing the canonical-related factor (e.g. p-Smad3) and non-canonical-related factors (e.g p-38; p-SAPK/JNK1/2 and p-Erk1/2), C2C12 cells treated with myostatin, a factor known to result in muscle atrophy through canonical and noncanonical pathways, have shown increased response to myostatin in myoblasts, not much in myotubes, indicating that this response is dependent on differentiation stage of C2C12 cells. Reduced response of myotubes to myostatin could be attributed to the increased expression of follistatin, a known blocker of myostatin receptor—ACVR2. Co-culture the C2C12 with tumor cell C26, which expresses myostatin, increased inflammatory signals but not canonical and non-canonical signals. However, this phenomenon cannot be recapitulated in human skeletal muscle CHQ cells, indicating the existence of cell-type specific characteristic. Due to the non-canonical pathway is related to inflammatory response, inflammatory-related factors are also examined and showed an elevated response (p-STAT3) in myotubes. These responses are majority compared at 2-hour treatment time point.

Comments:

  1. The writing is wordy and rather hard to read. There are some mistakes in the sentences, for example, in the introduction, line 56 “small mother???” makes no sense.

  1. The fond size is not equal throughout the figures. Some figures are very small, such as Figure 1; 2 3, 7 and 8.

  1. Lack of deep elaboration. For example, the peak responses are shown at 2-hour time point, there is no explanation for why this time point and what cause the difference compare to other time points.

  1. Why the tumorkine (tumor-derived cytokines) secretion was evaluated after 24-hour co-culture instead of 2.

  1. Lack of loading control. Although there is stain free blot, it’s better to show whether the loading control is equal or not, such as GAPDH/Tubulin/Actin, etc…

  1. The conclusion claimed that murine C2C12 cells are not recommended cell line to study the effects of myostatin/ACVR2 due to the inconsistent results compare to human muscle cell line. There are fundamental differences between mouse and human, the difference should be expected. I don’t think this comparison is reasonable.

Reviewer 2 Report

Here, Lautaoja et al. present a manuscript entitled "Differentiation of the murine C2C12 myoblasts 2 strongly reduces their responsiveness to myostatin 3".
The manuscript is well written and contains only very low number of typos. But I really think that this manuscript requires significant improvements.
First, interestingly a number of studies with telling titles investigated in the past the role of myostatin using immortalized murine C2C12 muscle cells. Rodgers et al. published in "Endocrinology" (2014) that "Myostatin stimulates, not inihibits, C2C12 myoblast proliferation". Uemura et al. published in "FEBS Open Bio." (2017) that "Myostatin promotes tenogenic differentiation of C2C12 myoblast cells through Smad3". Langley et al. published in "JBC" (2002) that "Myostatin Inhibits Myoblast Differentiation by Down Regulating MyoD Expression". Perie et al. published in "BBR" (2016) that "Enhancement of C2C12 myoblast proliferation and differentiation by GASP-2, a myostatin inhibitor". Graham et al. published in "BBR" (2017) that "Recombinant myostatin reduces highly expressed microRNAs in differentiating C2C12 cells". Suprisingly, only the paper of Graham et al. is mentioned in this study. To obtain a more complete picture of what is known and how to relate the finding of the authors to previous reports, the authors should really discuss their findings with what is known and already reported!
Second, in this study murine immortalized C2C12 cells were compared with CHQ muscle cells; the latter are in fact human primary myogenic cells from healthy donors. This is a little like comparing apples to pears. Why not compare murine primary muscle satellite cells with CHQ ? I want to see the authors adressing the effect of myostatin using mouse primary muscle cells!
Third, the quality of the figures is terrible due to non-acceptible resolution. This might have happened during online submission by the "biomolecules" homepage. To whom it may concern: It is necessary to ensure good quality figures for reviewers.
Fourth, there are some discrepancies in the figures. per example:
a. how can it be that 2h and 24h non-treated myoblasts have different amount of Smad3 (Fig. 1F)?
b. how helpful is Fig. 4E for understanding? If 2h and 24h samples show significantly different amount of marker protein, like p-SAPK, then why to pool ?
c. why is there only 2-fold more Follistatin protein in C2 MT (Fig. 6) compared with 20-fold more mRNA (Fig. 5)? Is it because translation is less efficient or because the zurnover time of Follistatin is high in MT ? Please provide experimental data.
Fifth, I am little bit confused about the impact of this manuscript. If the aim was to find an in vitro model for investigating the effects of myostatin on skeletal muscle, the authors now know that they can use CHQ cells ? Would it be not helpful to provide a little bit more of experimental data regarding myostatin effects using the CHQ cells ?

Reviewer 3 Report

The manuscript reports signaling responses to myostatin during the proliferation and differentiation stages in C2C12 and CHQ myoblast cultures. Both the canonical (involving Smad3) and non-canonical (involving MAPKs) signaling pathways of myostatin were examined. The effect of co-culture of colon cancer cells (C26 cells), expressing ‘tumorkine’ and C2C12 cells was also examined on signaling pathways in proliferating and differentiating C2C12 cells. The materials &methods are adequately described and results are in general clearly summarized. However, the interpretation of the results is not firmed supported by the data presented along with numerous unclear statements

Major comments:

It was stated that ‘The results demonstrate that the C2C12 cells lost most of their 22 canonical and non-canonical responsiveness to myostatin after differentiation’. The data, however, shows that the canonical pathway (Smad phosphorylation) and noncanonical pathway involving p38 were responded in C2C12 myotubes even though the SAPK/JNK and ERK pathways were not responded (Fig. 1). The authors appear to claim that unlike C2C12 cells, CHQ cells were responsive to myostatin regardless of their proliferation and differentiation stages. In CHQ cells, the ERK and SAPK.JNK phosphorylation was not measured in response to myostatin (Fig. 7), so no idea whether those two pathways were activated in response to myostatin. In this regard, it appears that no difference in the responsiveness to myostatin exists between C2C12 and CHQ myotubes.

The ‘responsiveness to myostatin’ used multiple time in the manuscript, including in the title, appears to be a vague statement since it is not clearly defined the type of response. Thus, the conclusion, ‘C2C12 myotubes seem to be non-optimal, if not unsuitable, as an in vitro model for investigating the effects of myostatin on skeletal muscle,’, is not well justified.

Minor comments:

Line 29, ‘effects of myostatin on skeletal muscle’ – unclear and too broad

Line 53 – citations 16 and 17 are not closely related to the statement

Line 124 – The study used only the mature form of myostatin (12.5 kD), then it is questioned why CHO-S cells were co-transfected with cDNAs of myostatin propeptide and mature form of myostatin.

Line 234 – change ‘shown as’ into ‘as shown bythe’

Line 240 – change ‘in less extent myotube’ into ‘less extent in myotube’

Line 304 – change ‘yoblst’ to ‘myoblast’

Line 331-334, ‘These results ------- follistatin’ – vague and unclear statement. Also, appears to overstretch the logic. What do ‘metabolic difference’ and ‘myostatin metabolism’ mean?

Line 342-342 – as stated previously, the statement is not well justified

Line 362, ‘These observations highlight the vital role of follistatin in the skeletal muscle functionality and contractility’ – This statement appears to be an overstretch. No change in follistatin expression was observed in CHQ cells during differentiation (Fig. 8), so the overexpression of follistatin during differentiation is probably limited to C2C12 cells. In this regard, the role of follistatin in skeletal muscle functionality can not be generalized.

Line 375-377, ‘C2C12 myotubes are not a suitable model to investigate the effects of myostatin, possibly due to the endogenous regulation of TGF-β family members and their regulators during differentiation into myotubes ‘  - Again, not well justified statement. Needs to clearly define what kind effects of myostatin.

Line 454-456 – awkward wording